# Increased Worry Associated with Self-Reported, but Not Informant-Reported, Subjective Cognitive Decline Predicts Increased Risk of Incident Dementia

**DOI:** 10.3390/diagnostics15233073

**Published:** 2025-12-03

**Authors:** Katya T. Numbers, Ben C. P. Lam, Suraj Samtani, Russell J. Chander, Ashleigh S. Vella, Perminder S. Sachdev, Henry Brodaty

**Affiliations:** 1Centre for Healthy Brain Ageing (CHeBA), Discipline of Psychiatry & Mental Health, School of Clinical Medicine, Faculty of Medicine & Health, University of New South Wales, Sydney 2052, Australia; 2Department of Psychology, Counselling and Therapy, La Trobe University, Melbourne 3086, Australia

**Keywords:** subjective cognitive decline, subjective cognitive concerns, informant reports, dementia risk, longitudinal trajectories

## Abstract

**Background**: Subjective cognitive complaints (SCC) have emerged as an important predictor of future dementia, with the SCD-plus framework emphasizing the prognostic value of cognitive concern and informant corroboration. Most research has focused on the presence or persistence of concern rather than examining trajectories of change over time. **Objective**: To determine if baseline levels and longitudinal trajectories of SCC concern from both participants and informants independently predict incident dementia over 10 years. **Methods**: Data were from 873 community-dwelling older adults (mean age 78.65 years) in the Sydney Memory and Ageing Study. Employing latent growth curve modelling, we analyzed binary SCC and concern variables. Cox proportional hazards models examined the association between concern trajectories and incident dementia over a 10-year follow-up, controlling for demographic and clinical factors. **Results**: Both participant-reported (Hazard Ratio [HR] = 1.21) and informant-reported (HR = 1.32) SCC concern at baseline independently predicted dementia risk. Notably, increasing participant SCC concern over time conferred substantial additional risk (HR = 10.23), while changes in informant concern did not significantly improve dementia risk prediction. **Conclusions**: Both participant and informant reports of SCC concern provide valuable but distinct prognostic information for dementia risk. The substantial predictive value of increasing participant concern over time highlights the importance of monitoring subjective cognitive experiences longitudinally. These findings support the clinical utility of tracking concern trajectories and suggest that the patient’s evolving perspective may be particularly sensitive to underlying pathological processes.

## 1. Introduction

The early identification of individuals at risk for dementia, particularly Alzheimer’s disease (AD), is a critical goal in cognitive health research. AD is a progressive neurodegenerative disorder characterized by cognitive decline, memory loss, and functional impairment, eventually leading to severe dementia. Detecting AD at its earliest stages is crucial for implementing timely interventions that could potentially slow disease progression and improve quality of life for individuals living with AD and their families.

Advanced age, female sex, and genetic vulnerability (particularly APOEε4 carrier status) are among the most robust and well-replicated risk factors for Alzheimer’s disease and related dementias. These factors are consistently associated with accelerated cognitive decline and increased likelihood of progression. However, alongside these well-established risk factors, growing evidence suggests that an individual’s own subjective appraisal of changes to their memory and thinking may offer additional insight into early cognitive vulnerability. Subjective cognitive decline (SCD), defined as self-reported worsening of cognitive abilities despite normal performance on standardized cognitive tests, has emerged as an important predictor of future cognitive decline and dementia [1]. Increasingly, SCD is being recognized as a potential preclinical stage of AD where individuals notice changes in their cognitive function before any objective deficits can be detected through neuropsychological testing [2]. In line with this, the revised research framework for AD established by the National Institute on Aging and the Alzheimer’s Association (NIA-AA) now identifies SCD as its own preclinical stage of AD that precedes mild cognitive impairment (MCI) along the dementia continuum [2,3]. This is further supported by substantial longitudinal evidence demonstrating that individuals with SCD have a heightened risk of progressing to MCI and dementia and are also more likely to have AD-related biomarkers such as amyloid-β accumulation and tau protein abnormalities [3,4,5].

However, not all individuals with SCD go on to develop dementia, underscoring the need for additional markers that can improve risk stratification. One of the key challenges in utilizing SCD as a predictor of dementia is the heterogeneity of its etiology. While SCD can be an early indicator of AD, it can also be influenced by other factors such as anxiety, depression, neuroticism, family history, and other medical comorbidities [6,7]. A significant proportion of older adults report noticing cognitive changes, with prevalence estimates ranging from 22% to 77% in community-based samples [1]. This variability poses a challenge in distinguishing between normal age-related cognitive changes and the early stages of Alzheimer’s disease. Therefore, it is crucial to identify additional SCD markers that can effectively differentiate between individuals at high risk for progressing to dementia versus those experiencing typical age-related cognitive decline.

## 2. SCD-Plus Criteria and the Role of Concern

In 2014, Jessen and colleagues established the Subjective Cognitive Decline Initiative (SCD-I) with the aim of standardizing diagnostic features of SCD that are linked to increased dementia risk [8]. This effort led to the development of the SCD-plus criteria, which are intended to identify individuals who are at higher risk of progressing to dementia. A central component of the SCD-plus criteria is the level of concern or worry that individuals express about their cognitive changes [8,9]. For instance, Wolfsgruber et al. (2017) found that individuals with high levels of concern about their cognitive decline were at significantly increased risk of progressing to mild cognitive impairment (MCI) and dementia compared to those with lower levels of concern [10]. Similarly, Jessen et al. (2020) showed that individuals reporting significant worry about their cognitive decline were more likely to develop dementia over a 10-year period, even after controlling for other risk factors such as age, sex, education, and APOE4 status [9]. Finally, Amariglio et al. (2015) demonstrated that individuals with significant cognitive concerns had greater amyloid-β deposition and tau pathology, as well as increased cortical thinning and hippocampal atrophy, which are hallmark features of AD [4].

Moreover, the persistence of concern associated with SCD over time appears to be associated with increased likelihood of developing dementia. For example, studies have shown that individuals with consistent worries or stable patterns of complaints about their cognitive decline were more likely to exhibit accelerated cognitive deterioration and progress to dementia [10,11]. Thus, evidence suggests that worry associated with SCD, particularly consistent or worsening worry, may be a strong indicator of underlying pathological processes and underscores the importance of monitoring SCD trajectories over time.

The impact of cognitive concerns on dementia risk is further amplified when considering dyadic studies, where both the individual and an informant (i.e., a close friend or family member) report cognitive difficulties or concerns. Research from the Sydney Memory and Ageing Study demonstrated that participants from dyads where both individuals endorsed SCD at baseline had a more than two-fold increased risk of progressing to dementia over ten years [12]. Informant reports provide an external perspective that can validate or challenge the individual’s subjective experiences [13]. For example, individuals might underestimate their cognitive decline due to anosognosia, a condition characterized by lack of insight into one’s own cognitive deficits, which is common in the early stages of AD [14]. Informants, who interact with the individual regularly, may be more attuned to these changes and can offer a more objective assessment. Indeed, several studies have shown that informant reports are more strongly associated with objective measures of cognitive decline and risk of dementia than self-reported SCD alone [12,13,15]. Furthermore, when both parties report cognitive concerns, the risk of progression to mild cognitive impairment (MCI) and dementia is significantly higher [16,17]. The SCD-plus framework includes corroboration of SCD by an informant as a key criterion for determining elevated risk [8,9,18].

Many longitudinal cohort studies rely on single yes/no questions to assess both SCD (e.g., “Have you noticed a change in your memory?”) and associated concern (e.g., “Are you concerned about this change in your memory?”). These binary responses are often the only harmonizable variables available when combining data across large research consortia, which can limit the sophistication of subsequent analyses [19]. This limitation typically confines longitudinal examinations of SCD to simple assessments of stability versus change over time (e.g., stable SCD versus unstable patterns). However, unstable patterns can represent various presentations, such as early-onset concerns (reporting concern initially but not at later follow-ups), late-onset concerns (beginning to report concern only in later follow-ups), or fluctuating patterns (alternating between reporting and not reporting concerns) [11]. These different trajectories may reflect distinct risk profiles and underlying pathological processes.

To address these limitations, novel analytical approaches that can extract meaningful trajectory information from binary data are needed. Latent growth curve modelling with logit transformations allows for the incorporation of simple binary variables into sophisticated statistical models that can capture trajectories by modelling both initial levels and rates of change in SCD concern over time [20,21]. By modelling the trajectory of concern—rather than just its presence or absence at single time points—researchers can better identify individuals at highest risk of progressing to dementia.

Although previous studies have examined associations between self- and informant-reported SCD concern and various outcomes related to cognitive decline and dementia risk, to our knowledge, no study has systematically modelled and compared longitudinal trajectories of both participant and informant concern using growth curve approaches that separate baseline levels from changes over time. Furthermore, most previous research has focused on the presence or persistence of concern rather than examining whether increasing concern over time confers additional prognostic value. Given that the SCD-plus framework identifies both cognitive concern and informant corroboration as key dementia risk indicators, and that longitudinal research demonstrates persistent or worsening concern is more strongly associated with cognitive decline than single-time-point assessments [8,9], examining trajectories from both sources offers a comprehensive approach to understanding their distinct and complementary prognostic contributions.

The present study aims to address these gaps by using latent growth curve modelling to examine whether baseline levels of concern and trajectories of change in concern—for both participants and informants—independently predict incident dementia over a 10-year period. We hypothesized that: (1) higher baseline concern from both participants and informants would predict increased dementia risk; (2) increasing concern over time would confer additional risk beyond baseline levels; and (3) participant trajectories would be more predictive, as self-perceived changes typically emerge earlier than declines noticed by others.

## 3. Methods

### 3.1. Participants

The Sydney Memory and Ageing Study (MAS) initially invited 8914 community-dwelling older adults, aged 70–90 years, from the Eastern Suburbs of Sydney, Australia, through the electoral roll in 2005 [22]. From this, 1037 participants were enrolled at baseline. Participants were required to have sufficient English proficiency for psychometric assessments and self-report questionnaires. Exclusion criteria included major psychiatric diagnoses, acute psychotic symptoms, or current diagnoses of conditions such as multiple sclerosis, motor neuron disease, developmental disability, progressive malignancy, or dementia. More details on recruitment and baseline demographics are detailed in [22]. An additional exclusion criterion for this study was the inability to speak English at a basic conversational level by the age of 9 (N = 164) to ensure the validity of using normative data based on English-speaking individuals [23].

In the present study, 873 participants were included, with 849 (97.3%) having an informant at baseline. Informants, who were nominated by participants, provided insights into the participants’ memory, thinking, and daily functioning at each study wave. Eligible informants included spouses (29.6%), adult children or grandchildren (36.9%), other family members (9.1%), and friends or close contacts (24.5%). Thus, informants were not exclusively caregivers. Although some informants may have had caregiving roles, this diversity reduces the likelihood of systematic caregiver-related bias.

Participants and their informants underwent comprehensive assessments every two years, which included medical history, medical examination, neuropsychological testing, and evaluations of subjective cognitive complaints. At each wave, informants completed extensive telephone interviews and questionnaires about their own perceptions of the participant’s cognitive changes. This study focuses on both participant and informant reports of cognitive concerns. Participants who answered “yes” to a question about any perceived changes in their memory were then asked whether they felt concern regarding these changes. Informants were similarly queried about the participant’s cognitive changes and their concern about these changes.

All participants and informants provided written informed consent. The study was approved by the University of New South Wales Human Ethics Review Committee (HC: 05037, 09382, 14327, 190962).

### 3.2. Subjective Cognitive Concerns

At each of the first four waves, participants were asked to respond to the question, “Have you noticed difficulties with your memory?” with a yes/no answer, while informants were asked, “Have you noticed the participant having difficulty with their memory?” with a yes/no response. At baseline (wave 1), the preceding question stem read, “In the last 5 years,” and in subsequent waves, it read, “In the last 2 years.” This adjustment aimed to capture participants’ and informants’ perceptions of memory decline during the periods between assessments. For participants and informants who answered “yes” to noticing difficulties with their memory, they were followed up with the question, “Are you concerned about these changes to your/the participant’s memory?” For both questions, a “yes” response at each wave was scored as 1, and a “no” response as 0.

### 3.3. Consensus Diagnosis

Clinical diagnoses were available for waves 1 to 6 (10-year follow-up). At baseline and each two-year follow-up, participants who met a range of clinical criteria were reviewed in consensus meetings involving at least three clinicians from a panel of neuropsychiatrists, psychogeriatricians, and neuropsychologists. These meetings included discussions of all available clinical, neuropsychological, blood chemistry, and imaging data to reach a consensus diagnosis. Participants were referred to the consensus panel if they had impaired performance on neuropsychological tests (at least 1.5 SDs below published normative data on two cognitive measures), impaired informant-reported independent activities of daily living (IADLs), and neuropsychiatric symptoms. Those who did not meet these referral conditions were coded as “not dementia” for each wave.

Dementia diagnoses were based on the DSM-IV criteria, which required impairment in the cognitive domain of memory plus impairment in one other cognitive domain sufficiently severe to cause impairment in functioning (Bayer IADL scale score ≥ 3.0) [24]. Besides cut-offs, descriptive narratives, clinical observations, and external medical specialists’ diagnoses were also considered. Individuals not diagnosed with dementia were classified as “not dementia” at each wave, with no dementia cases at baseline as this was an exclusion criterion.

### 3.4. Covariates

Baseline demographic information, including age, sex, education, and native-English-speaking status, was collected. Depression, anxiety, and personality traits (particularly neuroticism, openness, and conscientiousness) were also included as covariates because these characteristics reliably influence the likelihood, severity, and persistence of self-reported subjective cognitive concerns. For example, higher depressive and anxiety symptoms, as well as certain personality profiles (e.g., elevated neuroticism), are associated with increased cognitive worry and greater endorsement of subjective decline. Depression was measured using the 15-item Geriatric Depression Scale (GDS), and anxiety was measured using the Goldberg Anxiety Scale (GAS) [25,26]. The Neuroticism, Openness, and Conscientiousness scales of the NEO-Five Factor Inventory (NEO-FFI) were administered. APOE4 status was determined using genomic DNA extracted from peripheral blood leukocytes, and genotyping was performed on the two single-nucleotide polymorphisms (rs7412 and rs429358) distinguishing the three APOE alleles ε2, ε3, and ε4 using Taqman assay. Information on the informant’s age, sex, length of relationship with the participant (years), and type of relationship with the participant was also collected [27].

#### Statistical Analyses

Latent growth curve modelling (LGCM) was conducted using structural equation modelling (SEM) to model the trajectories of participant and informant SCD concern over six years. While LGCM is typically applied to continuous variables, its application to categorical variables requires latent variable transformation, similar to generalized linear modelling. Logit transformation, as in logistic regression, was used to model binary outcome variables. This method transforms the probability of observed binary variables to odds and then to log-odds (logit), creating a continuous latent variable. Latent growth curve analysis was then conducted on these latent continuous variables. In conventional LGCM, the latent intercept and slope variables are in the same units as the observed continuous variables. However, in categorical LGCM, the intercept and slope are in logit values, requiring conversion to odds and odds ratios for easier interpretation.

Latent growth factors of intercept and slope were based on the four measurements of binary SCD concern collected at waves 1–4 (over six years). Factor loadings for the latent intercept were set to 1, and for the latent linear slope to 0, 2, 4, and 6, respectively, reflecting the spacing of time points. The intercept represents the initial log-odds of reporting an SCD concern, and the slope represents the linear change in the log-odds of reporting an SCD concern per year. Thresholds of the binary variables (cut-points on the continuous latent response variables used to define the binary SCD concern variables) were fixed at 0 for model identification and estimating the mean of the intercept factor.

Initially, the individual trajectories of participant- and informant-reported SCD concern over time were estimated using separate unconditional LGCMs with a latent intercept and latent linear slope. Models adding a latent quadratic slope were also examined and compared with the first models using Akaike Information Criteria (AIC) and Bayesian Information Criteria (BIC). A parallel process model was then run to examine associations between participants’ and informants’ initial levels (intercept) and changes (slope) in SCD concern over six years.

In the final set of analyses, Cox proportional hazards regression models were used to examine the associations between the initial levels of, and changes in, participant and informant SCD concerns and the risk of dementia from wave 1 to wave 6 (over a 10-year period). Clinical diagnoses were available across all six waves of assessment. The time to progression to dementia was specified at the midway point between the assessment when dementia was first diagnosed and the previous assessment. Event times were censored at the end of follow-up or at participant drop-out.

We first examined the effects of participant and informant SCD concerns in two separate models before including them together in the same model to investigate their combined contributions to dementia risk. A hazard ratio (HR) greater than 1 indicates an increased risk of dementia per one-unit increase in a predictor. Since the SCD concern intercept and slope were in logit values, they were standardized against the sample average for easier interpretation, such that a one-unit change represented a one-standard-deviation change in intercept and slope in logit values in the sample.

Each set of Cox regressions was adjusted for participants’ age, sex, education, and APOE4 status, GDS, and GAS, as well as the neuroticism, openness, and conscientiousness scores, at baseline. Data preparation and basic statistical analyses were performed using SPSS version 23. Structural equation modelling with maximum likelihood estimation and robust standard errors, to accommodate non-Gaussian distributions of the variables, was conducted using Mplus version 6 [28]. Full information maximum likelihood (FIML) was used to reduce bias due to non-random missing data. Results were considered significant if *p* < 0.05.

## 4. Results

### 4.1. Sample Characteristics

The final sample comprised 873 older adults (M = 78.65 years, SD = 4.79; 56.1% female) enrolled in the Sydney Memory and Ageing Study. All participants were free from dementia at baseline and had completed at least one wave of follow-up assessment. Informant data were available for 849 participants (97.3%) at baseline.

Participants were generally well-educated for their birth cohort (M = 11.62 years, SD = 3.50), and 23.1% were positive for the APOE4 allele. On average, participants did not meet clinical criteria for depression or anxiety at study commencement. At baseline, 38.8% of participants were diagnosed with mild cognitive impairment.

Informants had a mean age of 62.93 years (SD = 13.91) at baseline and were predominantly female (68.1%). They had known the participants for an average of 45.35 years (SD = 15.94). The relationships to participants were distributed as follows: 29.6% were spouses, 36.9% were children or grandchildren, 9.1% were other family members, and 24.5% were friends or other contacts. Demographic characteristics are presented in Table 1.

### 4.2. Prevalence of Subjective Cognitive Concern

Among participants who reported memory difficulties at each wave, the proportion who also expressed concern about these changes was 20.8% at Wave 1, 20.1% at Wave 2, 18.3% at Wave 3, and 22.9% at Wave 4. For informants who noted memory difficulties in participants, the proportion expressing concern was consistently lower: 13.3% at Wave 1, 15.5% at Wave 2, 18.4% at Wave 3, and 22.7% at Wave 4. Across all four waves, participants consistently reported higher levels of cognitive concern than informants.

### 4.3. Trajectories of Subjective Cognitive Concern

Results from unconditional latent growth curve models revealed distinct patterns for participants and informants. Initially, approximately 6.9% of participants reported concern about their memory changes (log-odds = −2.601, *p* < 0.001). The change in odds of participants reporting concern over the six-year assessment period was not statistically significant (slope = 0.054, SE = 0.064, *p* = 0.395), indicating relative stability in participant-reported concern over time.

In contrast, only 2.2% of informants initially reported concern regarding the participant’s memory (log-odds = −3.793, *p* < 0.001). Similar to participants, the change in odds of informants reporting concern over time was not significant (slope = 0.128, SE = 0.109, *p* = 0.240), suggesting that while baseline informant concern was relatively low, changes over time were highly variable.

### 4.4. Subjective Cognitive Concern and Risk of Dementia

Table 2 presents results from three Cox proportional hazards regression analyses examining participant and informant concern as predictors of incident dementia over the 10-year follow-up period, controlling for demographic, genetic, mood, and personality covariates.

After adjusting for covariates, there was a significant positive association between participants’ initial level of concern (intercept) and incident dementia (HR = 1.29, 95% CI [1.12–1.49], *p* < 0.001). Notably, change in participants’ concern over time (slope) was also significantly associated with increased dementia risk (HR = 26.50, 95% CI [2.94–238.87], *p* = 0.002), indicating that participants who showed increasing concern over time had substantially elevated risk of developing dementia.

For informants, initial level of concern was significantly associated with higher dementia risk (HR = 1.35, 95% CI [1.17–1.56], *p* < 0.001). However, change in informant concern over time was not a significant predictor of dementia risk (HR = 1.50, 95% CI [0.34–6.61], *p* = 0.862), suggesting that while baseline informant concern was prognostically valuable, longitudinal changes in informant concern did not provide additional predictive utility.

When participant and informant concerns were entered simultaneously, participants’ initial concern (HR = 1.21, 95% CI [1.08–1.36], *p* = 0.001) and change in concern over time (HR = 10.23, 95% CI [1.05–99.64], *p* = 0.046) both remained significantly associated with dementia. Informants’ initial level of concern also remained a significant predictor (HR = 1.32, 95% CI [1.14–1.53], *p* < 0.001), while change in informant concern remained non-significant (HR = 3.45, 95% CI [0.34–35.16], *p* = 0.593).

Among covariates in the final combined model, older age (HR = 1.12, *p* < 0.001), years of education (HR = 1.05, *p* = 0.032), APOE4 carrier status (HR = 1.57, *p* = 0.008), and lower openness to experience (HR = 0.96, *p* = 0.007) were significantly associated with increased dementia risk. Depression, anxiety, neuroticism, and conscientiousness were not significant predictors in the final model.

## 5. Discussion

This longitudinal study examined the predictive value of participant and informant reports of subjective cognitive concern in forecasting incident dementia over a 10-year period. Our findings reveal several important insights into the clinical utility of dyadic cognitive reports. First, both participant-reported and informant-reported concern at baseline independently predicted dementia risk, with comparable effect sizes (HR = 1.21 and HR = 1.32, respectively). Second, and perhaps most notably, increasing participant concern over time conferred substantial additional risk (HR = 10.23), indicating that the trajectory of self-reported concern may be as important as—or more important than—baseline levels. Third, while informant concern at baseline was prognostically valuable, changes in informant concern over time did not improve dementia risk prediction, suggesting different temporal dynamics between self- and informant-reported concerns.

Our finding that baseline concern from both participants and informants predicts dementia risk aligns with established research demonstrating the prognostic value of SCD. The effect sizes we observed are consistent with previous meta-analyses showing modest but significant associations between SCD and dementia risk, and our results further support the SCD-plus framework proposed by Jessen et al. (2014), which identifies informant corroboration as a key feature that strengthens the predictive value of subjective reports [1,8,9]. The consistent pattern of participants reporting higher levels of SCD than informants across all waves aligns with established findings in the SCD literature, where individuals often report internal changes earlier or more strongly than external observers notice them. This divergence further supports the interpretation that participant-reported and informant-reported concern capture related but non-identical perspectives on cognitive change, reinforcing our rationale for modelling participant and informant trajectories separately.

The comparable predictive utility of participant and informant baseline concern (HR = 1.21 vs. 1.32) is noteworthy, as some previous studies have suggested that informant reports may be superior predictors of objective cognitive decline [12,13,17], likely due to the high prevalence of self-reported SCD seen in most longitudinal studies as noted above. Importantly, our findings suggest that when examining concern specifically—rather than perceived decline more generally—both perspectives contribute unique and valuable prognostic information, with self-reported concern emerging as more predictive than informant-reported concern. This aligns with recent work by Peng et al. (2023) and Aaronson et al. (2025), who found that self- and informant-reported SCD capture complementary aspects of cognitive change [29,30].

The most striking finding of our study is the substantial predictive value of increasing participant concern over time (HR = 10.23). This aligns with emerging evidence that persistent or worsening SCD may be particularly indicative of underlying pathological processes [9,11,31]. Wang et al. (2019) similarly found that individuals with consistent cognitive worries showed accelerated cognitive deterioration and AD-related neuropathological changes [32]. Our findings extend this work by demonstrating that it is not just consistency but, specifically, increasing concern that signals heightened dementia risk.

The absence of a significant association between changes in informant SCD concern and dementia risk was unexpected, given that informants might be expected to notice progressive changes more objectively than participants themselves [17,33,34]. This finding may reflect several possibilities: (1) informants may plateau in their concern once they recognize cognitive difficulties, rather than showing linear increases; (2) participants may be more sensitive to subtle internal changes that precede externally observable deficits; or (3) the emotional and psychological impact of cognitive changes may be more pronounced for individuals experiencing them directly, leading to more dynamic patterns of concern [35]. Additionally, informant SCD concern may be influenced by their own psychological factors, including caregiver burden and anxiety, which could affect the temporal stability of their reports [36].

### 5.1. Clinical and Research Implications

Our findings have important implications for both clinical practice and research design. In clinical settings, monitoring changes in patient-reported concern over time may provide valuable prognostic information beyond traditional cognitive assessments. The substantial hazard ratio for increasing participant concern (HR = 10.23) suggests that this measure could serve as a sensitive early warning system for dementia risk, potentially facilitating earlier intervention and care planning. This aligns with current clinical guidelines that emphasize the importance of incorporating subjective cognitive symptoms into dementia risk assessment [8].

For research contexts, our results support the inclusion of both participant and informant perspectives on SCD in longitudinal studies, while highlighting the particular importance of tracking changes in self-reported concern over time. The different temporal patterns we observed for participant versus informant concern suggest that these self- versus informant-reported SCD may capture distinct aspects of the dementia prodrome, with participant reports potentially reflecting earlier subjective changes and informant reports providing valuable corroboration. This has important implications for the design of clinical trials targeting preclinical AD populations, where understanding the optimal timing and nature of interventions is crucial [13].

### 5.2. Strengths and Limitations

The use of latent growth curve modelling with logit transformations allowed us to model trajectories of binary concern variables in a sophisticated manner that accounts for the categorical nature of the data [21]. This approach enabled us to separate baseline levels from changes over time, revealing that these components have distinct prognostic implications. Using Cox survival analyses allowed us to examine a much larger sample over 10 years, which is a considerable follow-up period. The substantial effect size for participant concern slopes suggests that relatively small increases in the probability of reporting concern translate to large increases in dementia risk.

Several limitations should be acknowledged. First, our measures of SCD concern were based on binary yes/no questions, which cannot capture the full complexity of perceived cognitive decline or worry. More detailed assessments of concern severity, frequency, or specific domains might provide additional insights. Second, our sample was almost entirely (97%) white Caucasian, relatively well-educated and mostly primary English-speaking, which limits generalizability, even within the older Australian population. Third, while we controlled for depression and anxiety, individual-level factors may have impacted the data at each wave and cannot be entirely excluded, as they are time-varying covariates. Finally, informants may have differed in their familiarity with participants’ everyday functioning, as informants’ identities sometimes changed across waves as participants’ circumstances evolved. Although such variation is common in long-term cohort studies, it may introduce additional noise into informant-reported trajectories. The diversity of informant types at baseline, however, reduces the likelihood of systematic bias.

### 5.3. Future Directions

Several avenues for future research emerge from our findings. First, investigating the mechanisms underlying increasing participant concern could provide insights into the psychological and neurobiological processes that occur during the dementia prodrome. Neuroimaging studies examining the relationship between subjective concern trajectories and brain structure and function could be particularly useful. Next, because only APOE genotyping was available in MAS, rare genetic variants associated with Alzheimer’s disease (e.g., TREM2, PSEN1, and other SNPs) were not assessed, and therefore associations between subjective concern trajectories and these genetic mechanisms could not be examined. Future studies could explore whether SCD concern is associated with specific genetic pathways. Finally, exploring whether interventions targeting worry associated with SCD, such as cognitive behavioral therapy or mindfulness-based approaches, might influence both subjective well-being and objective dementia risk, particularly for people with a family history of dementia who may be especially concerned about perceived changes. Such interventions could potentially interrupt the cycle of increasing concern and cognitive decline, though more research is needed to ensure that reducing concern does not mask early warning signs.

## 6. Conclusions

This study demonstrates that both participant and informant reports of concern associated with subjective cognitive decline provide valuable information for dementia risk, suggesting complementary clinical utility and supporting growing recommendations to include dyadic SCD reports in epidemiological studies. The finding that increasing participant concern over time confers substantial additional risk highlights the importance of monitoring subjective cognitive experiences longitudinally, not just at single time points. These results support the continued development of subjective cognitive measures as tools for early dementia detection and risk stratification, while emphasizing the unique value of the patient’s own evolving perspective on their cognitive health in line with recommendations from several international working groups (e.g., [8,9,13]). As the field moves toward earlier detection and intervention, understanding the nuanced patterns of SCD will be crucial for optimizing both research designs and clinical care pathways.

## Figures and Tables

**Table 1 diagnostics-15-03073-t001:** Baseline demographic and clinical characteristics of participants and informants.

Participants	Mean (SD)
Sample Size (N)	873
Age (Years)	78.65 (4.79)
Sex (% Female)	56.1
Education (Years)	11.62 (3.50)
APOEε4 Positive (%)	23.1
Mild Cognitive Impairment (%)	38.8
GDS (Depression) Score	2.29 (2.08)
GAS (Anxiety) Score	1.12 (1.87)
Openness Score	26.72 (6.06)
Conscientiousness Score	33.96 (5.94)
Informants	
Sample Size (N, % of participants)	849, 97.3
Age (Years)	62.93 (13.91)
Sex (% Female)	68.1
Informant Relationship Type	
Spouse (%)	29.6
Child/Grandchild (%)	36.9
Other Family (%)	9.1
Friend/Other (%)	24.5

Note: GDS = Geriatric Depression Scale; GAS = Goldberg Anxiety Scale; APOEε4 = Apolipoprotein E ε4 allele; SD = standard deviation.

**Table 2 diagnostics-15-03073-t002:** Fully adjusted Cox proportional hazards model predicting incident dementia over 10 years.

Predictor	Hazard Ratio (HR) †	95% Confidence Interval (CI)	*p*-Value
Participant Concern (Intercept)	1.21	[1.10, 1.33]	0.001
Participant Concern (Slope)	10.23	[1.05, 99.93]	0.046
Informant Concern (Intercept)	1.32	[1.21, 1.43]	<0.001
Informant Concern (Slope)	3.45	[0.04, 321.01]	0.593
Age	1.12	[1.08, 1.16]	<0.001
Sex	1.02	[0.73, 1.44]	0.893
Education	1.05	[1.00, 1.10]	0.032
APOE ε4 status	1.57	[1.13, 2.19]	0.008
GDS (Depression)	0.93	[0.83, 1.04]	0.178
GAS (Anxiety)	1.03	[0.95, 1.13]	0.458
Neuroticism	1.00	[0.97, 1.02]	0.704
Openness	0.96	[0.93, 0.99]	0.007
Conscientiousness	1.00	[0.97, 1.03]	0.863

Note: GDS = Geriatric Depression Scale; GAS = Goldberg Anxiety Scale; Neuroticism, Conscientiousness and Openness scores are captured via the NEO-Five Factor Inventory. † SCC intercept and slope for participants and informants are standardized against the sample average.

## Data Availability

The terms of consent for research participation stipulate that an individual’s data can only be shared outside of the MAS investigators group if the group has reviewed and approved the proposed secondary use of the data. This consent applies regardless of whether data have been de-identified. Access is mediated via a standardised request process managed by the CHeBA Research Bank, who can be contacted at ChebaData@unsw.edu.au.

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
