# Peer review of "Increased Worry Associated with Self-Reported, but Not Informant-Reported, Subjective Cognitive Decline Predicts Increased Risk of Incident Dementia"

_diagnostics, 2025, doi:10.3390/diagnostics15233073_

Round 1

Reviewer 1 Report

Comments and Suggestions for Authors

In the manuscript, the authors reported a longitudinal study examining the effects of subjective cognitive decline (SCD) on incident dementia in a community-based sample. The study was well designed, both self-reported and informant-reported data of SCD were collected, and the analyses were controlled for covariates. The results of the study indicated that SCD predicted the risk for dementia, and self-reported and informant-reported SCD independently account for the variance of the outcome variables in 10 years. The findings have tremendous practical implications, and the manuscript will contribute to the literature on dementia. Please see below for detailed comments:

  1. It would be helpful to provide the information about how the hypotheses were developed in the first paragraph on p.4.
  2. For Hypothesis 3, the direction of the predicted relationship is not clear. Please describe how self-reported and informant-reported data would be different in predicted dementia risk.
  3. What the portion of the informants of older adults were their caregivers? Would it be possible that the informants nominated by the older adults were biased?
  4. The analyses were controlled for various variables. In addition to those common covariates, e.g., gender, SES, and age, it is expected to justify why depression, anxiety, and Big-five personality traits were included.
  5. Were there any differences in SCD between self-reported and informant-reported data?

Reviewer 2 Report

Comments and Suggestions for Authors

The manuscript is well-written and data analysis presented effectively correlate the occurrence of cognitive decline to increased risk of future dementia. Overall, the manuscript will be of interest to readers and clinicians trying to investigate the clinically relevant risk factors associated with dementia and AD.

  1. Authors could elaborate (in brief) about the role of known risk factors such as sex, age, Apoe and other genetic factors as elucidated in the literature in the Introduction section.  
  2. Were the levels of amyloid deposition and tau pathology in individuals with significant cognitive decline/concerns comparable across male and female participants? Does sex as a variable have a predominant role in dictating these phenotypes in individuals with cognitive decline complaints?
  3. Was there an age based cut off for this study? Were the studies specifically analyzing a cohort based on a particular age group. 
  4. Can the authors comment or draw any association between worsened cognitive decline in the participants and occurrence of other genetic risk factors such TREM2, PSEN1 or other SNPs thereof. 
